# CFPT: Empowering Time Series Forecasting through Cross-Frequency Interaction and Periodic-Aware Timestamp Modeling

Feifei Kou [1 2 3]   Jiahao Wang [1 2]   Lei Shi [4]   Yuhan Yao [1 2]   Yawen Li [5]   Suguo Zhu [6]   Zhongbao Zhang [* 1]   Junping Du [1]

## Abstract

Long-term time series forecasting has been widely studied, yet two aspects remain insufficiently explored: the interaction learning between different frequency components and the exploitation of periodic characteristics inherent in timestamps. To address the above issues, we propose **CFPT**, a novel method that empowering time series forecasting through **C**ross-**F**requency Interaction (CFI) and **P**eriodic-Aware **T**imestamp Modeling (PTM). To learn cross-frequency interactions, we design the CFI branch to process signals in frequency domain and captures their interactions through a feature fusion mechanism. Furthermore, to enhance prediction performance by leveraging timestamp periodicity, we develop the PTM branch which transforms timestamp sequences into 2D periodic tensors and utilizes 2D convolution to capture both intra-period dependencies and inter-period correlations of time series based on timestamp patterns. Extensive experiments on multiple real-world benchmarks demonstrate that CFPT achieves state-of-the-art performance in long-term forecasting tasks. The code is publicly available at this repository: https://github.com/BUPT-SN/CFPT.

*Corresponding Author [1]School of Computer Science (National Pilot Software Engineering School), Beijing University of Posts and Telecommunications, Beijing 100876, China [2]Key Laboratory of Trustworthy Distributed Computing and Service, Beijing University of Posts and Telecommunications, Beijing 100876, China [3]Guangxi Key Laboratory of Trusted Software, Guilin University of Electronic Technology, Guilin 541004, China [4]State Key Laboratory of Media Convergence and Communication, Communication University of China, Beijing 100024, China [5]School of Economics and Management, Beijing University of Posts and Telecommunications, Beijing 100876, China [6]College of Computer Science and Technology, Hangzhou Dianzi University, Hangzhou 310018, China. Correspondence to: Zhongbao Zhang <zhongbaozb@bupt.edu.cn>.

*Proceedings of the 42nd International Conference on Machine Learning*, Vancouver, Canada. PMLR 267, 2025. Copyright 2025 by the author(s).

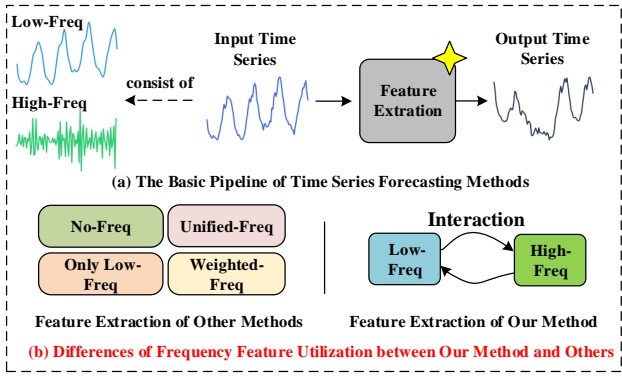

*Figure 1.* Comparison of frequency component utilization in time series forecasting.

## 1. Introduction

Temporal information data serves as one of the most fundamental and ubiquitous data types in real-world applications and intelligent systems (Liang et al., 2023; Hu et al., 2024; Liang et al., 2024a; Ilbert et al., 2024; Liang et al., 2024b). Within the broader domain of temporal data analysis, long-term time series forecasting has emerged as a particularly critical task with applications spanning numerous sectors, including energy consumption (Pinto et al., 2021), transportation (He et al., 2022), financial markets (He et al., 2023) and so on. Consequently, this task has garnered significant attention from researchers, leading to the development of numerous advanced methodologies (Nie et al., 2023; Yi et al., 2023; Zeng et al., 2023; Wang et al., 2024b; Lin et al., 2024; Luo & Wang, 2024; Xu et al., 2024a). Despite these advances, current research in long-term time series forecasting leaves two important aspects insufficiently explored.

First, as shown in Figure 1(a), time series data inherently contains signals of different frequencies with distinct characteristics, where low-frequency components carry fundamental patterns and trends, high-frequency components reflect short-term dynamics (Ye et al., 2024). The time series prediction task is mainly to predict the future time points by taking the historical time series as input and extracting their

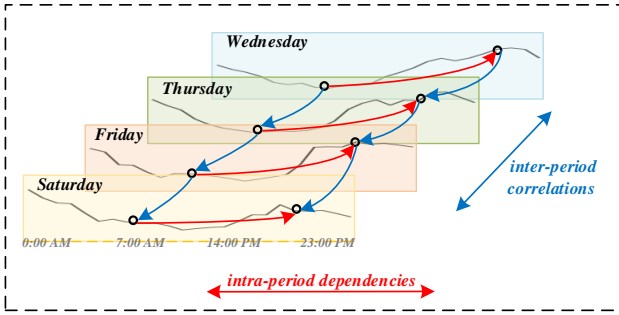

*Figure 2.* Illustration of intra-period dependencies and inter-period correlations in Electricity Consuming Load (ECL) dataset.

features. Based on the way frequency information is utilized during feature extraction, the existing methods can be classified into four categories as shown on the left side of Figure 1(b). The predominant approaches, classified as No-Freq Methods (Zhou et al., 2021; Liu et al., 2024a; Wang et al., 2024b), operate purely in the time domain and struggle to effectively process signals across the full frequency spectrum (Yi et al., 2024a). The other three categories attempt to address this limitation through frequency-domain processing: (1) Unified-Freq Methods (Yi et al., 2024b) that process frequency components but treat them uniformly without distinguishing their importance; (2) Only Low-Freq Methods (Zhou et al., 2022a; Xu et al., 2024b) that exclusively utilize low-frequency components while filtering out high-frequency ones; and (3) Weighted-Freq Methods (Zhou et al., 2022b; Zhang et al., 2024; Yi et al., 2024a) that combine high and low frequencies through simple weighted summation. Recent empirical studies (Ye et al., 2024; Zhang et al., 2024) reveal that the importance of different frequency components varies across scenarios, with each component potentially beneficial or detrimental to forecasting performance depending on the specific context, indicating that simply discarding certain frequency components or processing them independently may be suboptimal. However, although recent studies have introduced frequency-domain processing, they primarily focus on processing different frequency components independently, leaving the modeling of cross-frequency interactions unexplored.

Second, timestamps naturally reflect the periodic characteristics of time series data (Wang et al., 2024a). As shown in Figure 2, analysis of the data across four consecutive days (three weekdays and one weekend) in 24-hour periods reveals two distinct timestamp patterns. One is the intra-period dependencies, where each day exhibits similar periodic variations with time series values rising and falling at specific timestamps (as illustrated at the marked points 7:00 and 21:00). The other is the inter-period correlations, where load patterns at corresponding timestamps (e.g.,

values at 21:00) show strong consistency across weekdays while displaying notable differences between weekdays and weekends. Recent methods have demonstrated encouraging improvements by treating timestamps as enhancement components for time series forecasting (Wang et al., 2024a; Zeng et al., 2024). Some approaches attempt to model timestamps through feature embeddings (Zhou et al., 2021; Wu et al., 2021; 2023) or attention mechanisms (Liu et al., 2024a; Wang et al., 2024b), yet these methods have shown limited effectiveness in practice (Wang et al., 2024a). Despite these attempts in timestamp modeling, the periodic characteristics inherent in timestamps remain insufficiently explored. To fully harness the potential of timestamps for performance enhancement, exploring their periodic patterns becomes a promising direction.

In this paper, we propose **CFPT**, a novel framework that enhances long-term time series forecasting through two specialized branches. Firstly, We propose the Cross-Frequency Interaction (CFI) branch, it enables separate modeling of different frequency components while capturing their interactions through a carefully designed feature fusion mechanism. Thereby, It retains long-term evolutionary patterns and short-term dynamic features and providing a comprehensive representation across time scales. Secondly, we design the Periodic-aware Timestamp Modeling (PTM) branch. It transforms 1D timestamp sequences into 2D tensors based on fixed period lengths. Through 2D convolution operations, it captures both intra-period dependencies and inter-period correlations, enhancing the temporal context of the prediction and improving the prediction results.

Our main contributions can be summarized as follows:

- We propose CFPT, a novel dual-branch framework for long-term time series forecasting. The framework leverages both frequency dynamics and timestamp patterns through Cross-Frequency Interaction (CFI) and Periodic-aware Timestamp Modeling (PTM) branches to enhance prediction accuracy.

- To effectively model frequency information in time series data, we propose a CFI branch that captures interactions between different frequency components through a feature fusion mechanism for enhanced prediction performance.

- To enhance forecasting, we develop a PTM branch that transforms 1D timestamps into 2D tensors, capturing intra- and inter-period relations via 2D convolutions. Experiments on multiple real-world benchmarks show our method's superiority.

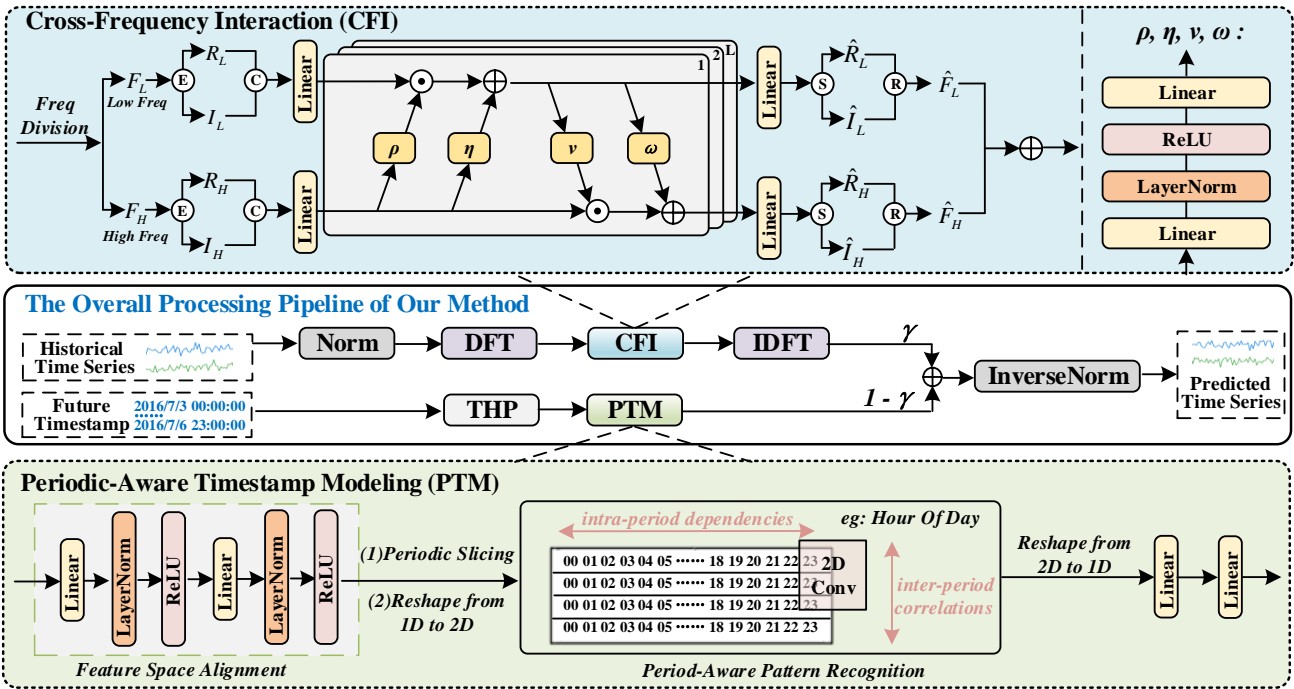

*Figure 3.* Overview of the proposed CFPT architecture. The middle part shows the overall processing pipeline, while the upper and lower parts illustrate the two main branches: Cross-Frequency Interaction (CFI) and Periodic-Aware Timestamp Modeling (PTM), respectively. Detailed meaning of the symbols: Ⓔ, extraction of real/imaginary parts; Ⓒ, concatenation; Ⓢ, splitting of processed features; Ⓡ, reconstruction of complex numbers.

## 2. Related Work

### 2.1. Frequency Modeling in Time Series Forecasting

Time series forecasting with frequency analysis has attracted increasing attention in deep learning research. Recent works have explored various frequency-based approaches to enhance model performance. As illustrated in Figure 1(b), These methods generally fall into four categories based on their frequency processing strategies: (1) No-Freq Methods (Zhou et al., 2021; Liu et al., 2024a; Wang et al., 2024b) that operate purely in the time domain, (2) Unified-Freq Methods (Yi et al., 2024a) that process all frequencies uniformly, (3) Only Low-Freq Methods (Zhou et al., 2022a; Xu et al., 2024b) that exclusively focus on low-frequency components, and (4) Weighted-Freq Methods (Zhou et al., 2022b; Zhang et al., 2024; Yi et al., 2024a) that combine frequencies through weighted summation. While these approaches have shown promising results, they either ignore frequency-domain information entirely or process different frequency components independently. Unlike these methods, we propose to explicitly model cross-frequency interactions, recognizing that the importance of different frequency components varies across scenarios and their interactions can significantly impact forecasting accuracy.

### 2.2. Timestamp Modeling in Time Series Forecasting

Timestamps have been increasingly utilized as enhancement components for time series forecasting (Wang et al., 2024a; Zeng et al., 2024). Early approaches like Informer (Zhou et al., 2021), Autoformer (Wu et al., 2021), FEDformer (Zhou et al., 2022b) and TimesNet (Wu et al., 2023) utilize timestamps by adding timestamp embeddings with data embeddings, while recent methods such as iTransformer (Liu et al., 2024a) and TimeXer (Wang et al., 2024b) embed timestamp features as tokens for attention mechanisms. Additionally, with the advancement of large language models in time series forecasting (Zhou et al., 2023; Jin et al., 2023; Cao et al., 2024; Pan et al., 2024), some works (Liu et al., 2024b) have attempted to model timestamps through prompts. However, empirical studies (Wang et al., 2024a; Tan et al., 2024) reveal that such timestamp modeling approaches demonstrate limited effectiveness in practice. Furthermore, despite these attempts in timestamp modeling, the periodic characteristics inherent in timestamps remain insufficiently explored. To fully harness the potential of timestamps for performance enhancement, we propose to explicitly model both intra-period dependencies and inter-period correlations of time series based on timestamp patterns.

## 3. Preliminaries

### 3.1. Problem Statement

Given multivariate time series data $X \in \mathbb{R}^{N \times T}$ with $N$ variables over $T$ time steps and corresponding timestamp information $\mathcal{T} \in \mathbb{R}^{M \times T}$, where $X_{:,t} \in \mathbb{R}^{N \times 1}$ and $\mathcal{T}_{:,t} \in \mathbb{R}^{M \times 1}$ denote the observations and timestamps at time step $t$, respectively. The forecasting task aims to predict future $\tau'$ steps $\hat{X}_{t+1:t+\tau'}$ based on historical observations $X_{t-\tau:t}$, accompanied by their corresponding future timestamps $\mathcal{T}_{t+1:t+\tau'}$ to enhance prediction performance. The forecasting process can be formulated as:

$$\hat{X}_{t+1:t+\tau'} = \mathcal{F}(X_{t-\tau:t}, \mathcal{T}_{t+1:t+\tau'}) \tag{1}$$

where $\mathcal{F}(\cdot)$ denotes the proposed forecaster.

### 3.2. Discrete Fourier Transform (DFT & IDFT)

Time series data can be viewed as discrete samples of continuous signals, which can be transformed into frequency domain for spectral analysis. Given a multivariate time series input $X \in \mathbb{R}^{N \times T}$, we apply discrete Fourier transform (DFT) independently to each channel $x^{(i)} \in \mathbb{R}^{1 \times T}$. For real-valued signals, we use the single-sided spectrum representation $F[k] \in \mathbb{C}^{1 \times (\frac{T}{2}+1)}$:

$$\text{DFT} : F[k] = \sum_{n=0}^{T-1} x[n] \cdot e^{-i2\pi \frac{kn}{T}} \tag{2}$$

where $k \in [0, \frac{T}{2}]$ is the frequency index and $n \in [0, T-1]$ is the time step index in the original signal. Once in frequency domain, the components $F[k]$ can be characterized by their magnitude and phase, calculated from their real part $\text{Re}(F[k]) \in \mathbb{R}^{1 \times (\frac{T}{2}+1)}$ and imaginary part $\text{Im}(F[k]) \in \mathbb{R}^{1 \times (\frac{T}{2}+1)}$:

$$\text{Magnitude} : A[k] = \sqrt{\text{Re}(F[k])^2 + \text{Im}(F[k])^2} \tag{3}$$

$$\text{Phase} : \theta[k] = \text{atan2}(\text{Im}(F[k]), \text{Re}(F[k])) \tag{4}$$

where atan2 is the two-argument arctangent function that determines the angle in all four quadrants. The real and imaginary parts of frequency components represent symmetric and asymmetric patterns respectively. After modifying these components to adjust signal characteristics, we can transform the signal back to time domain using IDFT:

$$\text{IDFT} : x[n] = \frac{1}{T} \sum_{k=0}^{T/2} F[k] \cdot e^{i2\pi \frac{kn}{T}} \tag{5}$$

where $n \in [0, T-1]$ represents the reconstructed time indices. These transformations enable precise control over signal properties while maintaining the fundamental periodic patterns. In our implementation, the transformations are performed using Fast Fourier Transform (FFT) with a computational complexity of $O(T \log T)$.

### 3.3. Instance Normalization (Norm & InverseNorm)

Time series data often exhibits non-stationarity, where statistical properties like mean and variance shift over time. This can degrade model performance when making predictions on future data with different distributions (Kim et al., 2021; Fan et al., 2023; Han et al., 2024; Fan et al., 2025). To address this issue, we apply instance normalization to both input time series $X$ and predicted values $\hat{X}$. For input normalization:

$$\text{Norm} : X_{t-\tau:t}^{norm} = \frac{X_{t-\tau:t} - \mu}{\sqrt{\sigma + \epsilon}} \tag{6}$$

where $\mu$ and $\sigma$ are the mean and standard deviation of the input window respectively, and $\epsilon$ is a small constant for numerical stability. After obtaining predictions, we apply inverse normalization:

$$\text{InverseNorm} : \hat{X}_{t+1:t+\tau'} = \hat{X}_{t+1:t+\tau'}^{norm} \times \sqrt{\sigma + \epsilon} + \mu \tag{7}$$

where $\hat{X}_{t+1:t+\tau'}^{norm}$ is the predicted normalized data. This normalization strategy helps maintain consistent statistical properties across different time periods, enabling more robust forecasting performance.

### 3.4. Timestamp Hierarchical Processing (THP)

To better serve periodic-aware timestamp modeling, we first process the raw timestamp information $\mathcal{T}_{t+1:t+\tau'} \in \mathbb{R}^{M \times \tau'}$ through a Timestamp Hierarchical Processing (THP) (Alexandrov et al., 2020) module that extracts inherent timestamp hierarchies. Specifically, THP is a feature engineering module that decomposes timestamps into hierarchical timestamp features at different granularities. Using the notation "X of Y" (written as XOfY) where X represents a finer time unit and Y represents its parent unit, we extract timestamp features including MinuteOfHour in [0, 59] for minute-level fluctuations, HourOfDay in [0, 23] for diurnal patterns, DayOfWeek in [0, 6] for weekly cycles, MonthOfYear in [0, 11] for monthly patterns, and SeasonOfYear in [0, 3] for quarterly patterns, among others. Each timestamp component is normalized to [-0.5, 0.5] through carefully designed transformations that preserve their cyclic characteristics. The processed hierarchical timestamp features are denoted as $\mathcal{T}_{t+1:t+\tau'}^{P} \in \mathbb{R}^{m \times \tau'}$, where $m$ indicates the number of selected timestamp features.

## 4. Methodology

### 4.1. Structure Overview

The middle part of Figure 3 shows the overall architecture of our model, which consists of two branches: Cross-Frequency Interaction (CFI) and Periodic-Aware Timestamp Modeling (PTM). Specifically, as shown in the upper part of Figure 3, the CFI branch first applies DFT to decompose

time series into high and low frequency components. The frequency components are then processed by extracting and concatenating their real and imaginary parts, followed by separate linear projections. A coupling layer with $L$ iterations enables interactions between different frequency bands. The interacted features are transformed back to complex domain and combined through inverse Fourier transform to generate frequency-aware predictions.

Meanwhile, as illustrated in the lower part of Figure 3, the PTM branch first processes raw timestamps through THP to obtain $m$ representative timestamp features from $M$ potential hierarchical features that capture patterns at different granularities. These features are then passed through a simple linear feature extraction module to align with the $N$ dimensional time series space. The aligned features are sliced and reshaped according to period length $P$, enabling 2D convolution to capture both intra-period dependencies (within each period) and inter-period correlations (across different periods) of time series based on timestamp patterns. Two dimension-preserving linear mappings further refine the timestamp representations.

Finally, outputs from both branches are combined through weighted addition. Based on the detailed description above of our two complementary branches and their integration, the complete pipeline of CPFT can be mathematically formulated as follows:

$$X_{t-\tau:t}^{norm} = \text{Norm}(X_{t-\tau:t}) \tag{8}$$

$$\mathcal{T}_{t+1:t+\tau'}^{P} = \text{THP}(\mathcal{T}_{t+1:t+\tau'}) \tag{9}$$

$$\hat{X}_{CFI} = \text{CFI}(X_{t-\tau:t}^{norm}) \tag{10}$$

$$\hat{X}_{PTM} = \text{PTM}(\mathcal{T}_{t+1:t+\tau'}^{P}) \tag{11}$$

$$\hat{X}_{t+1:t+\tau'}^{norm} = \gamma \cdot \hat{X}_{CFI} + (1-\gamma) \cdot \hat{X}_{PTM} \tag{12}$$

$$\hat{X}_{t+1:t+\tau'} = \text{InverseNorm}(\hat{X}_{t+1:t+\tau'}^{norm}) \tag{13}$$

where $\gamma \in (0,1)$ serves as a constant weight coefficient. This dual-branch design leverages both frequency dynamics and timestamp patterns to enhance prediction accuracy.

### 4.2. Cross-Frequency Interaction (CFI)

To effectively capture and leverage both long-term trends and short-term fluctuations in time series forecasting, we propose the Cross-Frequency Interaction (CFI) branch that explicitly models the interactions between high-frequency and low-frequency components.

**Frequency Division.** First, we transform the normalized input $X_{t-\tau:t}^{norm} \in \mathbb{R}^{N \times \tau}$ into frequency domain through DFT, yielding $F \in \mathbb{R}^{N \times \omega}$, where $\omega = \frac{\tau}{2} + 1$. Using the median frequency as threshold, we separate $F$ into low-frequency components $F_L \in \mathbb{R}^{N \times \omega}$ and high-frequency components $F_H \in \mathbb{R}^{N \times \omega}$ through complementary binary masks.

**Initial Feature Processing.** The real and imaginary parts of $F_L$ and $F_H$ are extracted as $R_L, I_L \in \mathbb{R}^{N \times \omega}$ and $R_H, I_H \in \mathbb{R}^{N \times \omega}$ respectively, and then concatenated to obtain $g_L, g_H \in \mathbb{R}^{N \times 2\omega}$ before passing through separate linear layers, producing feature representations $h_L \in \mathbb{R}^{N \times D}$ and $h_H \in \mathbb{R}^{N \times D}$, where $D$ is the hidden dimension. This separation allows independent processing of symmetric and asymmetric patterns while preserving their mathematical relationships.

**Cross-Frequency Interaction.** Given the feature representations $h_L$ and $h_H$, a coupling network with $L$ layers is designed to model their interactions. Each coupling layer contains four MLPs with identical architecture ($\rho$, $\eta$, $\nu$, and $\omega$) that transform the features through:

$$h_L^{k+1} = h_L^k \odot \exp(\rho(h_H^k)) + \eta(h_H^k) \tag{14}$$

$$h_H^{k+1} = h_H^k \odot \exp(\nu(h_L^{k+1})) + \omega(h_L^{k+1}) \tag{15}$$

where $\odot$ denotes element-wise multiplication, and $k$ indicates the layer index. The multiplicative terms modulate the magnitude of frequency components while preserving their phase relationships, and the additive terms enable adjustments to both symmetric and asymmetric patterns. This coupling mechanism allows comprehensive feature interaction between different frequency bands, facilitating the modeling of how long-term trends influence short-term variations and vice versa. The resulting representations $\hat{h}_L$, $\hat{h}_H \in \mathbb{R}^{N \times D}$ are then projected through linear layers to obtain $\hat{g}_L, \hat{g}_H \in \mathbb{R}^{N \times 2w'}$, where $w' = \frac{\tau'}{2} + 1$.

**Frequency Recombination.** Finally, the processed features $\hat{g}_L$ and $\hat{g}_H$ are split to obtain the real and imaginary parts for each frequency component. Specifically, we split $\hat{g}_L \in \mathbb{R}^{N \times 2w'}$ to obtain low-frequency components $\hat{R}_L, \hat{I}_L \in \mathbb{R}^{N \times w'}$ and split $\hat{g}_H \in \mathbb{R}^{N \times 2w'}$ to obtain high-frequency components $\hat{R}_H, \hat{I}_H \in \mathbb{R}^{N \times w'}$, where the first $w'$ features represent the real parts and the remaining $w'$ features represent the imaginary parts. This splitting operation exactly reverses the concatenation performed in the initial feature processing stage, ensuring mathematical consistency when reconstructing complex numbers for the IDFT process. These parts are then used to reconstruct complex numbers $\hat{F}_L \in \mathbb{R}^{N \times w'}$ and $\hat{F}_H \in \mathbb{R}^{N \times w'}$ respectively. These components are summed and transformed back through IDFT to obtain the prediction $\hat{X}_{CFI} \in \mathbb{R}^{N \times \tau'}$.

### 4.3. Periodic-Aware Timestamp Modeling (PTM)

While the CFI branch effectively captures frequency-domain interactions, timestamp patterns and periodicities in timestamps also provide crucial information for forecasting. To further enhance the prediction accuracy, we propose the Periodic-Aware Timestamp Modeling (PTM) branch that hierarchically processes timestamp information and leverages 2D convolution to capture both intra-period dependencies

and inter-period correlations.

**Feature Space Alignment.** After obtaining the processed hierarchical timestamp features $\mathcal{T}_{t+1:t+\tau'}^{P} \in \mathbb{R}^{m \times \tau'}$, we need to align them with the dimensionality of the time series variables to enable effective timestamp pattern modeling. This alignment is achieved through a lightweight feature extraction module consisting of two sequential Linear-LayerNorm-ReLU blocks, which maps the $m$-dimensional timestamp features to the $N$-dimensional time series space. The aligned features $\mathcal{Z}_A \in \mathbb{R}^{N \times \tau'}$ retain the original sequence length while matching the variable dimension for subsequent periodic-aware modeling.

**Period-Aware Pattern Recognition.** To effectively capture periodic patterns in the aligned timestamp features $\mathcal{Z}_A$, we first reorganize them into a 2D structure according to the given period length $P$. Specifically, we reshape $\mathcal{Z}_A \in \mathbb{R}^{N \times \tau'}$ into $\hat{\mathcal{Z}}_A \in \mathbb{R}^{N \times P \times \lfloor \tau'/P \rfloor}$, where each slice along the last dimension represents one complete period. This transformation enables us to apply dimension-preserving 2D convolution operations with kernel size $k \times k$ that can simultaneously model both intra-period dependencies and inter-period correlations. The 2D convolution structure facilitates the modeling of complex periodic patterns in timestamp features by capturing both local periodic structures and their long-range relationships. After the convolution operations, we employ two dimension-preserving linear mappings along the variable and timestamp dimensions respectively to further refine these learned periodic representations. Finally, we obtain $\hat{X}_{PTM} \in \mathbb{R}^{N \times \tau'}$ to enhance the prediction performance through subsequent integration.

### 4.4. Dual-Branch Integration

The CFI and PTM branches are designed to capture complementary aspects of time series patterns. While the CFI branch focuses on modeling frequency-domain interactions between different frequency components, the PTM branch specializes in extracting periodic patterns from timestamp dependencies. As shown in Equation (12), these complementary features are integrated through weighted addition with a constant coefficient $\gamma$, allowing the model to effectively combine frequency-domain knowledge with timestamp pattern awareness. This dual-branch design creates a synergistic effect where the strengths of both frequency-based decomposition and periodic-aware timestamp modeling are leveraged to enhance the overall prediction accuracy.

### 4.5. Optimization Objective

For our forecasting task, we employ the squared loss (L2) to measure the discrepancy between the prediction and the ground truth. The overall training objective is:

$$\mathcal{L} = \|\hat{X}_{t+1:t+\tau'} - X_{t+1:t+\tau'}\|_2^2 \qquad (16)$$

where $\hat{X}_{t+1:t+\tau'}$ is computed following Equation (13) and $X_{t+1:t+\tau'}$ represents the ground truth values.

## 5. Experiments

In this section, we conduct extensive experiments on real-world time series datasets to evaluate the effectiveness of our proposed CFPT. We first introduce the experimental setup, followed by comprehensive performance comparisons with state-of-the-art methods. Furthermore, we perform detailed ablation studies on both CFI and PTM branches, analyze hyperparameter sensitivity and computational efficiency, and visualize prediction results to demonstrate the effectiveness of our framework.

### 5.1. Experimental Setup

**Dataset.** We conduct long-term forecasting experiments on seven popular real-world benchmarks including the ETT series (Zhou et al., 2021), ECL (Wu et al., 2021), Traffic (Wu et al., 2021), and Weather (Wu et al., 2021). These datasets span multiple domains including electricity transformer data, power consumption patterns, highway traffic monitoring, and meteorological measurements. To maintain consistency and enable direct performance comparison, we adopt identical data preprocessing steps (e.g.,TimesNet (Wu et al., 2023), iTransformer (Liu et al., 2024a)) to ensure fair comparison. The detailed statistics of these datasets are summarized in Appendix A.

**Baselines.** We compare our proposed CFPT with representative and state-of-the-art models, including: (1) TimeXer (Wang et al., 2024b), a parallel multivariate forecasting model leveraging exogenous variables; (2) FilterNet (Yi et al., 2024a), a signal processing-based model utilizing frequency filters; (3) iTransformer (Liu et al., 2024a), which treats series as variate tokens to capture multivariate correlations; (4) PatchTST (Nie et al., 2023), which extracts local patterns through subseries patches with channel independence; (5) FEDformer (Zhou et al., 2022b), implementing sparse attention in frequency domain; (6) TimesNet (Wu et al., 2023), capturing multi-scale temporal dependencies; (7) DLinear (Zeng et al., 2023) and (8) RLinear (Li et al., 2023), utilizing linear mappings for forecasting.

**Implementation Details.** Our experiments are performed with PyTorch 2.0.0 (Paszke et al., 2019) on a single NVIDIA RTX 3090 GPU. For forecasting task, we choose an input sequence length of $\tau = 96$ and evaluate on varying forecasting horizons $\tau' \in \{96, 192, 336, 720\}$. For timestamp features, we consider MinuteOfHour, HourOfDay, DayOfWeek, MonthOfYear, and SeasonOfYear in the Timestamp Hierarchical Processing (THP). And for the period length $P$ in the periodic timestamp slicing of PTM branch, we empirically set it to 24 to maintain a consistent gran-

ularity across all datasets. We adopt the mean absolute error (MAE) and the mean square error (MSE) as evaluation metrics. Additional implementation details are provided in Appendix B.

## 5.2. Main Results

Table 1 shows the strong performance of CFPT in long-term time series forecasting. Compared with forecasters of different structures, CFPT achieves robust performance across various datasets. Although not achieving the best results on the Traffic dataset, CFPT demonstrates superior generalization ability on other benchmarks, validating its effectiveness in handling diverse time series patterns.

From the frequency modeling perspective, models like TimeXer (Wang et al., 2024b), iTransformer (Liu et al., 2024a), and RLinear (Li et al., 2023) that operate solely in the time domain show limited capability in capturing complex frequency patterns. While FilterNet (Yi et al., 2024a) and FEDformer (Zhou et al., 2022b) attempt to process different frequency components through weighted summation, they fail to model the intricate interactions between frequency bands. Taking the ETT series as an example, where both high-frequency fluctuations and low-frequency trends coexist, our cross-frequency interaction mechanism effectively captures the dynamic relationships between different frequency components. This advantage is particularly evident on the ECL dataset, which exhibits rich periodic patterns at multiple frequency scales due to daily and weekly electricity consumption cycles, leading to more accurate predictions.

From the timestamp modeling perspective, methods such as DLinear (Zeng et al., 2023) and FilterNet (Yi et al., 2024a) that do not utilize timestamp information show limitations in capturing timestamp patterns. While TimesNet (Wu et al., 2023) incorporates timestamps through simple addition with positional encodings, and TimeXer (Wang et al., 2024b) and iTransformer (Liu et al., 2024a) treat timestamps as attention tokens, these approaches cannot fully exploit the inherent periodic characteristics of time series. This is particularly crucial for the ECL dataset, which contains complex temporal dependencies due to its regular daily patterns overlaid with weekly variations and seasonal trends. Our periodic-aware timestamp modeling explicitly captures both intra-period dependencies and inter-period correlations, enabling better understanding of these hierarchical timestamp patterns. The experimental results on ECL validate the effectiveness of our timestamp modeling approach.

The superior performance across different benchmarks demonstrates that our dual-branch architecture, which combines frequency interaction and periodic-aware timestamp modeling, provides a more comprehensive solution for long-term time series forecasting.

## 5.3. Ablation Study

**Using Frequency Division.** In this section, we experiment with the effect of frequency decomposition by comparing CFPT with its variant without separating high- and low-frequency components (w/o CFI-D). As shown in Table 2, the average results demonstrate that removing frequency decomposition consistently leads to performance degradation, indicating the importance of modeling frequency components separately.

**Using Cross-Frequency Interaction.** In this section, we study the effectiveness of the cross-frequency interaction by comparing CFPT with its variant without couple layers (w/o CFI-C). As shown in Table 2, removing the coupling layers leads to consistent performance degradation across all datasets, with particularly notable drops on ETTh2 (from 0.364 to 0.394 in MSE). These results demonstrate the importance of modeling interactions between different frequency components.

**Using CFI Branch.** In this section, we investigate the necessity of frequency-based modeling by completely removing the CFI branch (w/o CFI). As shown in Table 2, the model performance significantly degrades without frequency analysis, with MSE increasing from 0.374 to 0.391 on ETTm1, 0.364 to 0.390 on ETTh2, and 0.240 to 0.251 on Weather. The substantial performance drop validates the effectiveness of our frequency-based design in CFPT.

**Using Future Timestamps.** In this section, we compare CFPT with its variant using historical timestamps (CFPT-HT), where the final linear layer in PTM branch maps from historical to future sequence length instead of directly modeling future timestamps. As shown in Figure 4, the performance gap between these two models widens as the prediction length increases from 96 to 720 steps, with CFPT consistently outperforming CFPT-HT (0.499 vs 0.544 in MSE at 720 steps). This suggests that using future timestamps becomes increasingly crucial for longer horizons, likely due to the information loss when mapping historical patterns to extended forecasting windows.

**Using 2D Timestamp Modeling.** In this section, we compare CFPT with its variant using 1D convolution without periodic slicing (CFPT-1DT). As shown in Table 3, implementing 2D timestamp modeling consistently yields better performance across all datasets, with MSE improving from 0.376 to 0.374 on ETTm1, 0.369 to 0.364 on ETTh2, and 0.242 to 0.240 on Weather. These results demonstrate the advantage of our period-based 2D modeling strategy.

**Using PTM Branch.** In this section, we study the effectiveness of timestamp modeling by completely removing the PTM branch. As shown in Table 3, removing timestamp modeling leads to significant performance degradation

*Table 1.* Forecasting results with look-back window $\tau = 96$ and prediction lengths $\tau' \in \{96, 192, 336, 720\}$. The MSE and MAE metrics are averaged across all prediction horizons, with lower values indicating better performance. The best results are shown in bold. Our full results are in Appendix D.

| Methods | CFPT | | TimeXer | | FilterNet | | iTransformer | | PatchTST | | FEDformer | | TimesNet | | DLinear | | RLinear | |
|---|---|---|---|---|---|---|---|---|---|---|---|---|---|---|---|---|---|---|
| Metrics | MSE | MAE | MSE | MAE | MSE | MAE | MSE | MAE | MSE | MAE | MSE | MAE | MSE | MAE | MSE | MAE | MSE | MAE |
| ETTm1 | **0.374** | **0.393** | 0.382 | 0.397 | 0.384 | 0.398 | 0.407 | 0.410 | 0.387 | 0.400 | 0.448 | 0.452 | 0.400 | 0.406 | 0.403 | 0.407 | 0.414 | 0.407 |
| ETTm2 | **0.269** | **0.315** | 0.274 | 0.322 | 0.276 | 0.322 | 0.288 | 0.332 | 0.281 | 0.326 | 0.305 | 0.349 | 0.291 | 0.333 | 0.350 | 0.401 | 0.286 | 0.327 |
| ETTh1 | **0.433** | **0.429** | 0.437 | 0.437 | 0.440 | 0.432 | 0.454 | 0.448 | 0.469 | 0.454 | 0.440 | 0.460 | 0.458 | 0.450 | 0.456 | 0.452 | 0.446 | 0.434 |
| ETTh2 | **0.364** | **0.393** | 0.367 | 0.396 | 0.378 | 0.404 | 0.383 | 0.407 | 0.387 | 0.407 | 0.437 | 0.449 | 0.414 | 0.427 | 0.559 | 0.515 | 0.374 | 0.398 |
| ECL | **0.164** | **0.259** | 0.171 | 0.270 | 0.201 | 0.285 | 0.178 | 0.270 | 0.205 | 0.290 | 0.214 | 0.327 | 0.192 | 0.295 | 0.212 | 0.300 | 0.219 | 0.298 |
| Traffic | 0.470 | 0.289 | 0.466 | 0.287 | 0.521 | 0.340 | **0.428** | **0.282** | 0.481 | 0.304 | 0.610 | 0.376 | 0.620 | 0.336 | 0.625 | 0.383 | 0.626 | 0.378 |
| Weather | **0.240** | **0.267** | 0.241 | 0.271 | 0.248 | 0.274 | 0.258 | 0.278 | 0.259 | 0.281 | 0.309 | 0.360 | 0.259 | 0.287 | 0.265 | 0.317 | 0.272 | 0.291 |
| 1st Count | **6** | **6** | 0 | 0 | 0 | 0 | 1 | 1 | 0 | 0 | 0 | 0 | 0 | 0 | 0 | 0 | 0 | 0 |

*Table 2.* Ablation study on frequency modeling. Results are averaged from all forecasting horizons.

| Methods | CFPT | | w/o CFI-D | | w/o CFI-C | | w/o CFI | |
|---|---|---|---|---|---|---|---|---|
| Metrics | MSE | MAE | MSE | MAE | MSE | MAE | MSE | MAE |
| ETTm1 | **0.374** | **0.393** | 0.379 | 0.394 | 0.391 | 0.398 | 0.391 | 0.398 |
| ETTh2 | **0.364** | **0.393** | 0.369 | 0.397 | 0.394 | 0.410 | 0.390 | 0.407 |
| Weather | **0.240** | **0.267** | 0.242 | 0.269 | 0.250 | 0.275 | 0.251 | 0.276 |

*Table 3.* Ablation study on timestamp modeling. Results are averaged from all forecasting horizons.

| Methods | CFPT | | CFPT-1DT | | w/o PTM | |
|---|---|---|---|---|---|---|
| Metrics | MSE | MAE | MSE | MAE | MSE | MAE |
| ETTm1 | **0.374** | **0.393** | 0.376 | 0.394 | 0.385 | 0.395 |
| ETTh2 | **0.364** | **0.393** | 0.369 | 0.398 | 0.370 | 0.396 |
| Weather | **0.240** | **0.267** | 0.242 | 0.269 | 0.255 | 0.276 |

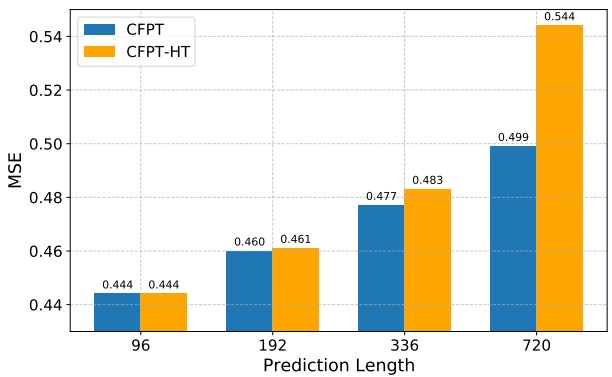

*Figure 4.* MSE comparison between CFPT (using future timestamps) and CFPT-HT (using history timestamps) on Traffic dataset.

across all datasets, with MSE increasing from 0.374 to 0.385 on ETTm1, 0.364 to 0.370 on ETTh2, and 0.240 to 0.255 on Weather. These results validate the importance of explicit timestamp modeling in our framework.

### 5.4. Hyperparameter and Efficiency Analysis

We analyze CFPT's hyperparameter sensitivity by examining critical parameters including weight coefficient $\gamma$, kernel size $k$, number of coupling layers $L$, and model dimension $D$, as well as evaluate its computational efficiency in terms of training time and resource consumption. CFPT achieves robust performance while maintaining high computational

efficiency. Detailed analyses on hyperparameter sensitivity and computational efficiency are referred to Appendix C.

### 5.5. Visualization of Prediction Performance

We present a prediction showcase on the ECL dataset, as shown in Figure 5. We select TimeXer (Wang et al., 2024b), FilterNet (Yi et al., 2024a), and iTransformer (Liu et al., 2024a) as the representative comparison methods. Comparing with these state-of-the-art models, we observe that CFPT delivers more accurate predictions of future series variations, demonstrating superior forecasting performance.

### 6. Conclusion

In this paper, we propose CFPT, a novel framework that empowers long-term time series forecasting through cross-frequency interaction and periodic-aware timestamp modeling. The Cross-Frequency Interaction (CFI) branch and Periodic-Aware Timestamp Modeling (PTM) branch work collaboratively, where CFI explicitly models interactions between different frequency components through a dedicated coupling mechanism, while PTM enhances prediction by capturing both intra-period dependencies and inter-period correlations based on timestamp patterns via 2D convolution on period-based tensors. Our comprehensive experiments across multiple benchmarks demonstrate the superiority of CFPT, achieving robust performance especially on datasets

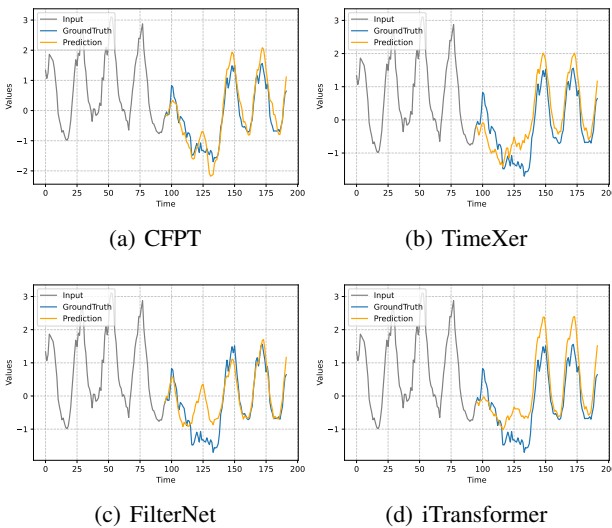

*Figure 5.* Visualization of prediction on the ECL dataset with lookback and horizon length as 96.

with complex periodic patterns like ECL. For future work, incorporating adaptive period detection mechanisms could further enhance our PTM branch to better accommodate datasets with irregular periodicity patterns. We hope our work can inspire future research to explore more sophisticated frequency interaction mechanisms and timestamp modeling strategies for enhancing long-term time series forecasting capabilities.

## Acknowledgements

This work was supported by the National Key Research and Development Program of China (2023YFF0725103), the National Natural Science Foundation of China (No.62002027, 62472042, 62072488, 62272058, U23A20319), the 8th Young Elite Scientists Sponsorship Program by CAST (2022QNRC001), Guangxi Key Laboratory of Trusted Software (No.KX202304), and the Beijing Natural Science Foundation (Grant No.L233034).

## Impact statement

This paper presents work whose goal is to advance the field of Machine Learning. There are many potential societal consequences of our work, none which we feel must be specifically highlighted here.

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

## A. Dataset Descriptions

We evaluate CFPT on seven widely-used time series benchmarks: ETT series (Zhou et al., 2021) (ETTh1, ETTh2, ETTm1, ETTm2), ECL (Wu et al., 2021), Traffic (Wu et al., 2021), and Weather (Wu et al., 2021). The ETT series contains transformer temperature records with different sampling rates (hourly and 15-minute). ECL consists of electricity consumption data from multiple clients, Traffic records road occupancy rates from highway sensors, and Weather includes various meteorological measurements. These datasets cover diverse scenarios with different sampling frequencies and variable dimensions, providing a comprehensive testbed for long-term forecasting. The detailed statistics of these datasets are summarized in Table 4.

*Table 4.* Statistical Properties of Experimental Datasets.

| Dataset | ETTm1&2 | ETTh1&2 | ECL | Traffic | Weather |
|---|---|---|---|---|---|
| Timesteps | 69,680 | 17,420 | 26,304 | 17,544 | 52,696 |
| Channels | 7 | 7 | 321 | 862 | 21 |
| Dataset Size | (34465, 11521, 11521) | (8545, 2881, 2881) | (18317, 2633, 5261) | (12185, 1757, 3509) | (36792, 5271, 10540) |
| Granularity | 15 mins | 1 hour | 1 hour | 1 hour | 10 mins |
| Periodicity | Daily | Daily | Daily & Weekly | Daily & Weekly | Daily & Seasonly |

## B. Implementation Details

All experiments are implemented in PyTorch 2.0.0 (Paszke et al., 2019) and conducted on a single NVIDIA RTX 3090 GPU. We utilize ADAM optimizer (Kingma, 2014) with learning rate searched from $\{0.01, 0.005, 0.0001, 0.0005\}$ and L2 loss for model optimization. The training process is fixed to 10 epochs with batch size selected from $\{4, 8, 16, 128\}$. For model architecture, we conduct grid search over several key hyperparameters: (a) Weight coefficient $\gamma$ from $\{0.1, 0.2, ..., 0.9\}$, (b) Kernel size $k$ from $\{2, 3, 4, 5\}$, (c) Number of coupling layers $L$ from $\{1, 3, 6, 12\}$, and (d) Model dimension $D$ from $\{128, 256, 512\}$. For timestamp features, we consider MinuteOfHour, HourOfDay, DayOfWeek, MonthOfYear, and SeasonOfYear in the Timestamp Hierarchical Processing (THP). Additionally, a fixed period length $P = 24$ is applied in PTM branch. All experiments are repeated with seed 2025. We reproduced all baseline models following the official implementation of TimesNet (Wu et al., 2023).

## C. Model Analysis

### C.1. Hyperparameter Sensitivity Analysis

To evaluate the parameter sensitivity of CFPT, we conduct ablation studies by varying key hyperparameters: (a) Weight coefficient $\gamma$ from $\{0.1, 0.2, ..., 0.9\}$, (b) Kernel size $k$ from $\{2, 3, 4, 5\}$, (c) Number of coupling layers $L$ from $\{1, 3, 6, 12\}$, and (d) Model dimension $D$ from $\{128, 256, 512\}$. The results on ETTm1, ETTh2 and Weather datasets with lookback length of 96 and horizon length of 720 are illustrated in Figure 6. The model shows optimal performance when $\gamma$ is in [0.1, 0.7], with relatively stable or degraded performance beyond 0.7. A kernel size of $k = 2$ generally yields better results. The impact of coupling layers $L$ varies across datasets, which may be attributed to their different frequency characteristics. The model achieves optimal performance with hidden dimension $D = 512$.

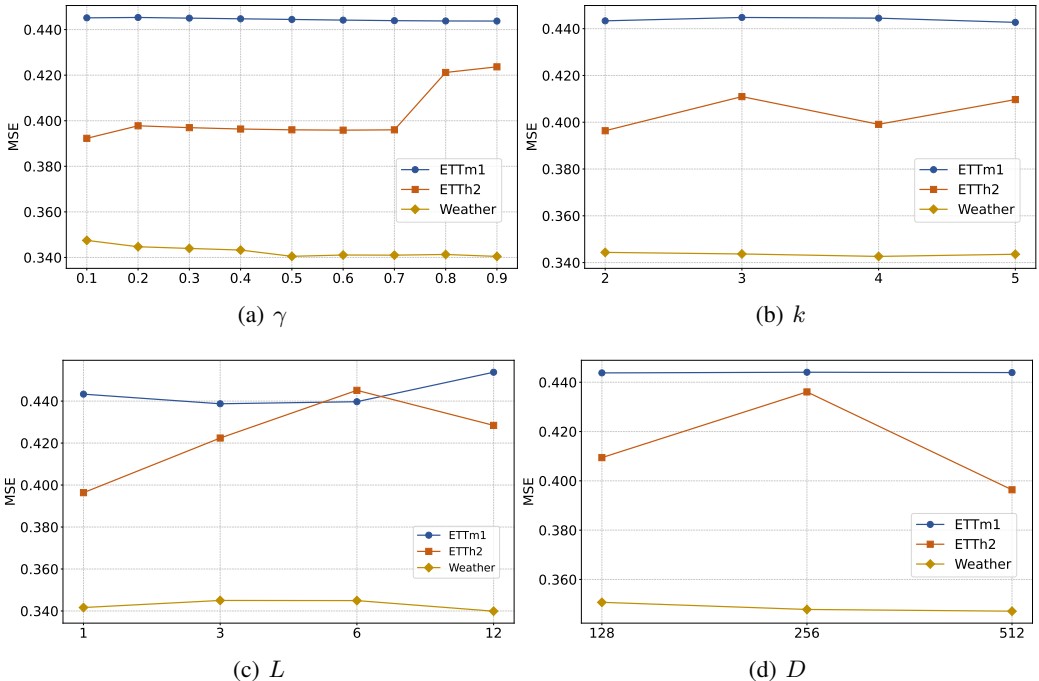

*Figure 6.* Sensitivity analysis of hyperparameters on ETTm1, ETTh2 and Weather datasets with lookback length of 96 and horizon length of 720.

## C.2. Efficiency Analysis

As shown in Table 5, we evaluate CFPT's efficiency based on model parameters, GPU memory usage, and training time per iteration on ETTm1, ETTh2 and Weather datasets with lookback window size fixed as 96 while varying prediction horizons from 96 to 720 timesteps. The results demonstrate that our model maintains stable training speed around 0.8s per iteration and exhibits low resource requirements with moderate parameter sizes and GPU memory usage across all settings, even for the Weather dataset with higher dimensionality and longer prediction horizons. These results indicate that CFPT achieves strong performance with modest computational costs, making it suitable for practical deployment.

*Table 5.* Model efficiency analysis with fixed lookback window size of 96 and varying prediction horizons, evaluated on different datasets in terms of model parameters, GPU memory usage, and training time.

| Dataset | Horizon | Parameter (MB) | GPU Memory (MiB) | Running Time (s / iter) |
|---|---|---|---|---|
| ETTm1 | 96 | 48.971 | 75.675 | 0.813 |
| | 192 | 4.744 | 15.982 | 0.804 |
| | 336 | 1.879 | 12.780 | 0.805 |
| | 720 | 7.631 | 20.341 | 0.831 |
| ETTh2 | 96 | 1.243 | 10.925 | 0.781 |
| | 192 | 17.344 | 33.619 | 0.789 |
| | 336 | 25.460 | 190.327 | 0.819 |
| | 720 | 21.262 | 52.163 | 0.789 |
| Weather | 96 | 12.507 | 248.786 | 0.793 |
| | 192 | 3.474 | 148.505 | 0.812 |
| | 336 | 3.924 | 167.071 | 0.800 |
| | 720 | 3.246 | 169.071 | 0.804 |

## D. Performance of Long-term Multivariate Forecasting

To comprehensively evaluate CFPT, we conduct long-term multivariate forecasting experiments on various real-world benchmarks. With lookback window size fixed as 96, we test the model's performance across different prediction horizons ranging from {96, 192, 336, 720}. The detailed results are presented in Table 6.

*Table 6.* Full results of the long-term multivariate forecasting task.

| Methods | | CFPT | | TimeXer | | FilterNet | | iTransformer | | PatchTST | | FEDformer | | TimesNet | | DLinear | | RLinear | |
|---|---|---|---|---|---|---|---|---|---|---|---|---|---|---|---|---|---|---|---|
| Metrics | | MSE | MAE | MSE | MAE | MSE | MAE | MSE | MAE | MSE | MAE | MSE | MAE | MSE | MAE | MSE | MAE | MSE | MAE |
| ETTm1 | 96 | **0.316** | **0.356** | 0.318 | **0.356** | 0.318 | 0.358 | 0.334 | 0.368 | 0.329 | 0.367 | 0.379 | 0.419 | 0.338 | 0.375 | 0.345 | 0.372 | 0.355 | 0.376 |
| | 192 | **0.354** | **0.380** | 0.362 | 0.383 | 0.364 | 0.383 | 0.377 | 0.391 | 0.367 | 0.385 | 0.426 | 0.441 | 0.374 | 0.387 | 0.380 | 0.389 | 0.391 | 0.392 |
| | 336 | **0.383** | **0.400** | 0.395 | 0.407 | 0.396 | 0.406 | 0.426 | 0.420 | 0.399 | 0.410 | 0.445 | 0.459 | 0.410 | 0.411 | 0.413 | 0.413 | 0.424 | 0.415 |
| | 720 | **0.444** | **0.434** | 0.452 | 0.441 | 0.456 | 0.444 | 0.491 | 0.459 | 0.454 | 0.439 | 0.543 | 0.490 | 0.478 | 0.450 | 0.474 | 0.453 | 0.487 | 0.450 |
| | Avg. | **0.374** | **0.393** | 0.382 | 0.397 | 0.384 | 0.398 | 0.407 | 0.410 | 0.387 | 0.400 | 0.448 | 0.452 | 0.400 | 0.406 | 0.403 | 0.407 | 0.414 | 0.407 |
| ETTm2 | 96 | **0.167** | **0.249** | 0.171 | 0.256 | 0.174 | 0.257 | 0.180 | 0.264 | 0.175 | 0.259 | 0.203 | 0.287 | 0.187 | 0.267 | 0.193 | 0.292 | 0.182 | 0.265 |
| | 192 | **0.232** | **0.292** | 0.237 | 0.299 | 0.240 | 0.300 | 0.250 | 0.309 | 0.241 | 0.302 | 0.269 | 0.328 | 0.249 | 0.309 | 0.284 | 0.362 | 0.246 | 0.304 |
| | 336 | **0.290** | **0.331** | 0.296 | 0.338 | 0.297 | 0.339 | 0.311 | 0.348 | 0.305 | 0.343 | 0.325 | 0.366 | 0.321 | 0.351 | 0.369 | 0.427 | 0.307 | 0.342 |
| | 720 | **0.385** | **0.389** | 0.392 | 0.394 | 0.392 | 0.393 | 0.412 | 0.407 | 0.402 | 0.400 | 0.421 | 0.415 | 0.408 | 0.403 | 0.554 | 0.522 | 0.407 | 0.398 |
| | Avg. | **0.269** | **0.315** | 0.274 | 0.322 | 0.276 | 0.322 | 0.288 | 0.332 | 0.281 | 0.326 | 0.305 | 0.349 | 0.291 | 0.333 | 0.350 | 0.401 | 0.286 | 0.327 |
| ETTh1 | 96 | **0.372** | **0.391** | 0.382 | 0.403 | 0.375 | 0.394 | 0.386 | 0.405 | 0.414 | 0.419 | 0.376 | 0.419 | 0.384 | 0.402 | 0.386 | 0.400 | 0.386 | 0.395 |
| | 192 | 0.425 | **0.421** | 0.429 | 0.435 | 0.436 | 0.422 | 0.441 | 0.436 | 0.460 | 0.445 | **0.420** | 0.448 | 0.436 | 0.429 | 0.437 | 0.432 | 0.437 | 0.424 |
| | 336 | 0.467 | **0.442** | 0.468 | 0.448 | 0.476 | 0.443 | 0.487 | 0.458 | 0.501 | 0.466 | **0.459** | 0.465 | 0.491 | 0.469 | 0.481 | 0.459 | 0.479 | 0.446 |
| | 720 | **0.466** | **0.461** | 0.469 | **0.461** | 0.474 | 0.469 | 0.503 | 0.491 | 0.500 | 0.488 | 0.506 | 0.507 | 0.521 | 0.500 | 0.519 | 0.516 | 0.481 | 0.470 |
| | Avg. | **0.433** | **0.429** | 0.437 | 0.437 | 0.440 | 0.432 | 0.454 | 0.448 | 0.469 | 0.454 | 0.440 | 0.460 | 0.458 | 0.450 | 0.456 | 0.452 | 0.446 | 0.434 |
| ETTh2 | 96 | **0.285** | **0.336** | 0.286 | 0.338 | 0.292 | 0.343 | 0.297 | 0.349 | 0.302 | 0.348 | 0.358 | 0.397 | 0.340 | 0.374 | 0.333 | 0.387 | 0.288 | 0.338 |
| | 192 | **0.363** | **0.388** | **0.363** | 0.389 | 0.369 | 0.395 | 0.380 | 0.400 | 0.388 | 0.400 | 0.429 | 0.439 | 0.402 | 0.414 | 0.477 | 0.476 | 0.374 | 0.390 |
| | 336 | **0.413** | 0.426 | 0.414 | **0.423** | 0.420 | 0.432 | 0.428 | 0.432 | 0.426 | 0.433 | 0.496 | 0.487 | 0.452 | 0.452 | 0.594 | 0.541 | 0.415 | 0.426 |
| | 720 | **0.396** | **0.422** | 0.408 | 0.432 | 0.430 | 0.446 | 0.427 | 0.445 | 0.431 | 0.446 | 0.463 | 0.474 | 0.462 | 0.468 | 0.831 | 0.657 | 0.420 | 0.440 |
| | Avg. | **0.364** | **0.393** | 0.367 | 0.396 | 0.378 | 0.404 | 0.383 | 0.407 | 0.387 | 0.407 | 0.437 | 0.449 | 0.414 | 0.427 | 0.559 | 0.515 | 0.374 | 0.398 |
| ECL | 96 | **0.136** | **0.231** | 0.140 | 0.242 | 0.176 | 0.264 | 0.148 | 0.240 | 0.181 | 0.270 | 0.193 | 0.308 | 0.168 | 0.272 | 0.197 | 0.282 | 0.201 | 0.281 |
| | 192 | **0.153** | **0.246** | 0.157 | 0.256 | 0.185 | 0.270 | 0.162 | 0.253 | 0.188 | 0.274 | 0.201 | 0.315 | 0.184 | 0.289 | 0.196 | 0.285 | 0.201 | 0.283 |
| | 336 | **0.168** | **0.265** | 0.176 | 0.275 | 0.202 | 0.286 | 0.178 | 0.269 | 0.204 | 0.293 | 0.214 | 0.329 | 0.198 | 0.300 | 0.209 | 0.301 | 0.215 | 0.298 |
| | 720 | **0.199** | **0.293** | 0.211 | 0.306 | 0.242 | 0.319 | 0.225 | 0.317 | 0.246 | 0.324 | 0.246 | 0.355 | 0.220 | 0.320 | 0.245 | 0.333 | 0.257 | 0.331 |
| | Avg. | **0.164** | **0.259** | 0.171 | 0.270 | 0.201 | 0.285 | 0.178 | 0.270 | 0.205 | 0.290 | 0.214 | 0.327 | 0.192 | 0.295 | 0.212 | 0.300 | 0.219 | 0.298 |
| Traffic | 96 | 0.444 | 0.274 | 0.428 | 0.271 | 0.506 | 0.336 | **0.395** | **0.268** | 0.462 | 0.295 | 0.587 | 0.366 | 0.593 | 0.321 | 0.650 | 0.396 | 0.649 | 0.389 |
| | 192 | 0.460 | 0.280 | 0.448 | 0.282 | 0.508 | 0.333 | **0.417** | **0.276** | 0.466 | 0.296 | 0.604 | 0.373 | 0.617 | 0.336 | 0.598 | 0.370 | 0.601 | 0.366 |
| | 336 | 0.477 | 0.289 | 0.473 | 0.289 | 0.518 | 0.335 | **0.433** | **0.283** | 0.482 | 0.304 | 0.621 | 0.383 | 0.629 | 0.336 | 0.605 | 0.373 | 0.609 | 0.369 |
| | 720 | 0.499 | 0.313 | 0.516 | 0.307 | 0.553 | 0.354 | **0.467** | **0.302** | 0.514 | 0.322 | 0.626 | 0.382 | 0.640 | 0.350 | 0.645 | 0.394 | 0.647 | 0.387 |
| | Avg. | 0.470 | 0.289 | 0.466 | 0.287 | 0.521 | 0.340 | **0.428** | **0.282** | 0.481 | 0.304 | 0.610 | 0.376 | 0.620 | 0.336 | 0.625 | 0.383 | 0.626 | 0.378 |
| Weather | 96 | **0.154** | **0.200** | 0.157 | 0.205 | 0.164 | 0.210 | 0.174 | 0.214 | 0.177 | 0.218 | 0.217 | 0.296 | 0.172 | 0.220 | 0.196 | 0.255 | 0.192 | 0.232 |
| | 192 | **0.203** | **0.242** | 0.204 | 0.247 | 0.214 | 0.252 | 0.221 | 0.254 | 0.225 | 0.259 | 0.276 | 0.336 | 0.219 | 0.261 | 0.237 | 0.296 | 0.240 | 0.271 |
| | 336 | **0.261** | **0.286** | **0.261** | 0.290 | 0.268 | 0.293 | 0.278 | 0.296 | 0.278 | 0.297 | 0.339 | 0.380 | 0.280 | 0.306 | 0.283 | 0.335 | 0.292 | 0.307 |
| | 720 | **0.340** | **0.339** | **0.340** | 0.341 | 0.344 | 0.342 | 0.358 | 0.347 | 0.354 | 0.348 | 0.403 | 0.428 | 0.365 | 0.359 | 0.345 | 0.381 | 0.364 | 0.353 |
| | Avg. | **0.240** | **0.267** | 0.241 | 0.271 | 0.248 | 0.274 | 0.258 | 0.278 | 0.259 | 0.281 | 0.309 | 0.360 | 0.259 | 0.287 | 0.265 | 0.317 | 0.272 | 0.291 |
| 1st Count | | **29** | **30** | 3 | 3 | 0 | 0 | 4 | 4 | 0 | 0 | 2 | 0 | 0 | 0 | 0 | 0 | 0 | 0 |

