# OpenReview forum: "CFPT: Empowering Time Series Forecasting through Cross-Frequency Interaction and Periodic-Aware Timestamp Modeling"
_ICML.cc/2025/Conference — ICML 2025 poster_

### Official Review · Reviewer_7jYd · 2025-03-06

**Overall Recommendation:** 4

**Summary:**

The paper introduces CFPT method including two branches to address two key limitations in existing methods: inadequate modeling of interactions between different frequency components and insufficient exploitation of timestamp periodicity. The CFI branch processes signals in the frequency domain and captures interactions between different frequency components, and the PTM branch transforms timestamp sequences into 2D tensors to identify both intra-period and inter-period patterns. Experiments show that CFPT offers an effective solution for time series forecasting.

## update after rebuttal
I keep the overall recommendation of "4 accept", as the authors have addressed my major questions and concerns.

**Claims And Evidence:**

The claims are generally well-supported. The paper provides comprehensive experimental results across multiple datasets showing CFPT outperforms baseline models. Ablation studies effectively demonstrate the contribution of each component. The visualizations and efficiency analysis further strengthen the evidence for the model's effectiveness.

**Essential References Not Discussed:**

N/A

**Experimental Designs Or Analyses:**

Yes, I have checked the soundness of the experimental designs, and I believe they are well-designed for the time series forecasting task.

**Methods And Evaluation Criteria:**

The methods and evaluation aspects are well-designed for the time series forecasting task. The architecture provides a reasonable approach for handling both frequency and temporal patterns, while the evaluation follows common practices in time series prediction by using appropriate benchmarks across diverse domains with standard performance metrics.

**Other Comments Or Suggestions:**

N/A

**Other Strengths And Weaknesses:**

Strengths:
1)	The CFPT framework is a novelty. It can advance the long-term time series forecasting domain by considering both the complex interrelationships between frequency components and the periodic patterns in timestamps, providing a new perspective for addressing challenging forecasting tasks and inspiring future research on sophisticated frequency interaction mechanisms and timestamp modeling strategies.
2)	The paper demonstrates a well-structured presentation. It presents a logical progression from problem formulation to experimental validation, effectively communicating both theoretical concepts and implementation details.
3)	The experiments are reproducible, with detailed implementation information provided. The authors clearly describe all experimental details including software versions, hardware specifications, and hyperparameter settings.
4)	The method's superior performance across multiple tasks demonstrates its potential for practical application in real-world environments where accurate long-term forecasting is critical.

Weaknesses:
1.	It is encouraged to increase the font size in Figure 5 to improve readability.
2.	The conclusion lacks discussion of limitations, which would have provided a more balanced assessment of the work and potential directions for improvement.

**Questions For Authors:**

1)	Could you increase the font size in Figure 5 to improve readability?
2)	What do you consider to be the limitations of your approach? Understanding potential constraints would help contextualize your results and suggest directions for future work.

**Relation To Broader Scientific Literature:**

The paper makes significant contributions to the broader scientific literature on frequency-based modeling and timestamp-based modeling in time series forecasting.

**Theoretical Claims:**

N/A

---

> ### Author Rebuttal · Authors · 2025-03-31
>
> Many thanks for your review and precious comments and advises. Specific responses are presented below:
>
> **Response to Question 1:**
>
> Thank you for your valuable comments. We did notice that the text in Figure 5 appears too small. In the revised version, we will increase the font size and enhance the clarity of text in Figure 5 to ensure better readability for readers.
>
> **Response to Question 2:**
>
> Thank you for your valuable suggestion about discussing our approach's limitations. We appreciate the recommendation to include this important aspect in our paper. In the revised version, we will add the following to our conclusion section: "For future work, incorporating adaptive period detection mechanisms could further enhance our PTM branch to better accommodate datasets with irregular periodicity patterns." This addition acknowledges a limitation of our current approach while suggesting a clear direction for improvement.

---

### Official Review · Reviewer_PJn9 · 2025-03-07

**Overall Recommendation:** 4

**Summary:**

This paper addresses the challenge of long-term time series forecasting by introducing CFPT, a method that integrates frequency component analysis with timestamp pattern recognition. The key innovation lies in modeling cross-frequency interactions while simultaneously capturing periodic characteristics in timestamps. Experiments are conducted on multiple tasks, and the results prove the effectiveness of the proposed framework.

# After rebuttal:

The authors solve my concerns and I vote for acceptance.

**Claims And Evidence:**

The submission provides strong and convincing support for its claims. The model achieves superior performance across seven benchmarks compared to baseline models, particularly on complex periodic datasets, while ablation studies demonstrate clear performance degradation when removing key components.

**Essential References Not Discussed:**

No essential references appear to be missing from the paper's discussion. The references are up-to-date, and appropriately contextualize the paper's contributions within the current state of research.

**Experimental Designs Or Analyses:**

The experimental design is generally sound and follows standard practices in time series forecasting. The authors use the input length of 96 timesteps and evaluate on multiple prediction horizons across seven widely-used benchmark datasets. The comparison with eight recent baseline models and the use of standard metrics is appropriate. The ablation studies are particularly thorough, systematically isolating the contributions of each component. For example, removing the Cross-Frequency Interaction (CFI) branch results in significant performance degradation, highlighting the importance of modeling frequency interactions for accurate forecasting.

**Methods And Evaluation Criteria:**

The proposed methods make good sense for the time series forecasting problem. The approach addresses two insufficiently explored aspects in long-term forecasting: the interaction between different frequency components and the periodic characteristics inherent in timestamps. Moreover, the paper employs a well-designed evaluation strategy for time series forecasting. The choice of benchmark datasets captures diverse real-world scenarios. The evaluation includes meaningful comparisons with leading baselines and examines the effectiveness of different model components.

**Other Comments Or Suggestions:**

For consistency, consider renaming section 3.2 to "Discrete Fourier Transform (DFT & IDFT)" to align with section 3.3's naming pattern, as both discuss paired operations.

**Other Strengths And Weaknesses:**

Strengths:

1.	The quality of the work is commendable, as the paper not only identifies key limitations in current forecasting paradigms but provides a principled solution that aligns with the interplay between different frequency components and the inherent periodic patterns in timestamps.

2.	The paper is well-organized with clear logic and thorough experimental design.

3.	The implementation is practically valuable, maintaining computational efficiency while achieving state-of-the-art performance. The stable training speed and moderate resource requirements make it suitable for real-world deployment.

Weaknesses:

1.	The author should provide justification for choosing DFT over the more computationally efficient FFT algorithm, while the FFT could reduce computational complexity from O(n²) to O(n log n).

2.	The visualization in Figure 2 could be improved. Arrows should follow chronological order to better represent temporal dependencies in the time series data, while the current arrows point in the opposite direction.

**Questions For Authors:**

1.	Why does the paper use DFT instead of FFT for frequency analysis? Would using FFT affect the model's performance or implementation?

2.	Could you correct the visualization of inter-period correlations in Figure 2? The arrows should follow chronological order rather than pointing in the opposite direction to accurately reflect temporal dependencies.

**Relation To Broader Scientific Literature:**

The key contributions of this paper connect to several important developments in time series forecasting literature. First, while previous works like iTransformer process time series in time domain, and recent methods like Fedformer and FilterNet explore frequency-domain processing, CFPT innovatively models cross-frequency interactions through a coupling network rather than processing components independently or using simple weighted summation. Second, the paper improves timestamp modeling beyond current methods that rely on embeddings, attention mechanisms, or prompts. Its periodic-aware approach using 2D convolutions effectively captures both intra-period dependencies and inter-period correlations, providing a more comprehensive solution to timestamp modeling.

**Theoretical Claims:**

I have verified the correctness of the theoretical claims and their proofs in the paper. The formulations of the Discrete Fourier Transform (DFT) and its inverse (IDFT) in equations (2) and (5) are mathematically sound, correctly establishing the foundation for frequency analysis and reconstruction. Additionally, the instance normalization proofs in equations (6) and (7) properly demonstrate the basis for time series data normalization and inverse normalization.

---

> ### Author Rebuttal · Authors · 2025-03-31
>
> Many thanks for your review and precious comments and advises. Specific responses are presented below:
>
>
> **Response to Question 1:**
>
> Thank you for your valuable comment. We will clarify that we implement the DFT using FFT algorithms in our work. Specifically, we will add the following statement at the end of Section 3.2: "In our implementation, these transformations are performed using Fast Fourier Transform (FFT) algorithms with a computational complexity of O(T log T)." This clarification will help readers understand our actual implementation approach.
>
> **Response to Question 2:**
>
> Thank you for your valuable comments. We are so sorry that we have mistaken the arrows. In the revised version, we will update Figure 2 to make all arrows follow chronological order.
>
>
> **Response to Other Comments Or Suggestions:**
>
> Thank you for your valuable comments. In the revised version, we will rename Section 3.2 to "Discrete Fourier Transform (DFT & IDFT)" to align with Section 3.3.

---

### Official Review · Reviewer_LJCA · 2025-03-10

**Overall Recommendation:** 4

**Summary:**

This paper targets the problem of long-term time series forecasting with attention to frequency components and timestamp patterns. The proposed method is based on two observations: the importance of different frequency components varies across scenarios and their interactions may impact forecasting accuracy, and timestamps naturally reflect periodic characteristics that remain insufficiently explored in existing approaches. Then a dual-mechanism forecasting framework named CFPT is proposed to capture both cross-frequency interactions through a dedicated coupling mechanism and periodic timestamp patterns using 2D convolution on period-based representations. Experiments are conducted across seven real-world benchmark datasets with varying prediction horizons, and the results demonstrate the superiority of the proposed framework compared to state-of-the-art methods.

## update after rebuttal

All my concerns have been addressed. I find this work both novel and solid, so I’d be glad to keep my score.

**Claims And Evidence:**

The claims in the submission are well-supported by clear evidence. The comprehensive benchmark results across multiple datasets demonstrate strong model performance. The thorough ablation studies effectively validate the contribution of each model module.

**Essential References Not Discussed:**

No essential references appear to be missing from this paper.

**Experimental Designs Or Analyses:**

The experimental design is methodologically sound. The authors evaluated CFPT on seven diverse datasets (ETT series, ECL, Traffic, Weather) against eight state-of-the-art baselines using standard metrics (MSE and MAE). Multi-horizon testing (96, 192, 336, 720 steps) effectively assessed long-term forecasting capabilities. Ablation studies systematically isolated each component's contribution through carefully designed variants (w/o CFI-D, w/o CFI-C, w/o CFI, CFPT-HT, CFPT-1DT, w/o PTM), confirming the necessity of frequency division, cross-frequency coupling, future timestamps, and 2D periodic modeling. Hyperparameter sensitivity analysis and computational efficiency evaluations further supported the model's robustness and practicality.

**Methods And Evaluation Criteria:**

(1)The proposed methods are well-suited for long-term time series forecasting. The model's design thoughtfully handles frequency components and periodic patterns, effectively capturing both temporal trends and fluctuations through its specialized branches.
(2)The evaluation criteria are appropriate and comprehensive. The testing utilizes diverse real-world benchmarks from environmental monitoring, power systems, transportation, and meteorological domains. The evaluation employs standard metrics and multiple prediction horizons, providing solid validation through baseline comparisons.

**Other Comments Or Suggestions:**

In Section 3.4, there is incorrect usage of quotation marks where two right quotation marks appear instead of a proper pair of left and right quotation marks.

**Other Strengths And Weaknesses:**

Strengths:
S1. The paper presents an innovative dual-branch architecture that addresses two insufficiently explored aspects in time series forecasting: the interaction learning between different frequency components and the exploitation of periodic characteristics inherent in timestamps.
S2. The authors provide clear motivation by systematically identifying limitations in existing approaches, classifying frequency-based methods into four categories and demonstrating how current timestamp modeling techniques show limited effectiveness in practice.
S3. The experimental evaluation demonstrates exceptional rigor with comprehensive testing across seven diverse real-world benchmarks and thorough ablation studies that validate each component's contribution to the overall performance improvements.
S4. CFPT shows excellent potential for real-world applications in domains like energy consumption and transportation, maintaining competitive computational efficiency while delivering superior performance on datasets with complex periodic patterns.

Weaknesses:
W1. The layout of Figure 3 should be improved to enhance readability. For example, the text "Reshape from 2D to 1D" should be repositioned directly above its corresponding arrow for better visual flow.
W2. Some mathematical notations should be clarified. Parameters k and n in Section 3.2 and summation parameter S in Section 4.5 would benefit from clearer explanations, which would improve clarity and enhance understanding of the methodology.

**Questions For Authors:**

Q1. Could the authors improve the layout of Figure 3 to enhance readability?
Q2. Could the authors address the mathematical notation issues that have been outlined in the weaknesses section?

**Relation To Broader Scientific Literature:**

(1) The paper advances the broader scientific literature on time series forecasting through its novel treatment of frequency components. Previous approaches can be categorized into four types based on their frequency handling: No-Freq Methods like [R1] that operate purely in the time domain, Unified-Freq Methods like [R2] that process all frequencies uniformly, Only Low-Freq Methods like [R3] that exclusively focus on low-frequency components, and Weighted-Freq Methods like [R4] that combine frequencies through weighted summation. This paper introduces a novel cross-frequency interaction mechanism that explicitly models the relationships between different frequency bands, building on recent empirical findings that demonstrate how the importance of frequency components varies across different forecasting scenarios.
(2) The paper contributes to timestamp modeling in time series forecasting. Early approaches like [R1] and [R5] incorporated timestamps through basic embeddings, while recent methods have explored more complex approaches: some like [R6] and [R7] treat timestamps as attention tokens, and others like [R8] have attempted to model timestamps through prompts. However, empirical studies reveal that these approaches show limited effectiveness. This paper uncovers the potential of explicitly modeling the periodic characteristics inherent in timestamps through a 2D convolution architecture that captures both intra-period dependencies and inter-period correlations.

References:
[R1] Zhou et al., "Informer: Beyond efficient transformer for long sequence time-series forecasting", AAAI 2021
[R2] Yi et al., "Frequency-domain mlps are more effective learners in time series forecasting", NeurIPS 2024
[R3] Zhou et al., "Film: Frequency improved legendre memory model for long-term time series forecasting", NeurIPS 2022
[R4] Zhou et al., "FEDformer: Frequency enhanced decomposed transformer for long-term series forecasting", ICML 2022
[R5] Wu et al., "Autoformer: Decomposition transformers with auto-correlation for long-term series forecasting", NeurIPS 2021
[R6] Liu et al., "iTransformer: Inverted transformers are effective for time series forecasting", ICLR 2024
[R7] Wang et al., "TimeXer: Empowering transformers for time series forecasting with exogenous variables", arXiv 2024
[R8] Liu et al., "Autotimes: Autoregressive time series forecasters via large language models", arXiv 2024

**Theoretical Claims:**

The theoretical claims about frequency decomposition and reconstruction are correct. These claims are verified through Equations 2-5, showing that signal transformation between time and frequency domains is reversible and preserves information (via DFT and IDFT), frequency components can be described by magnitude and phase, and complex signals can be decomposed and reconstructed using real and imaginary parts. These results confirm the model's frequency domain processing validity.

---

> ### Author Rebuttal · Authors · 2025-03-31
>
> Many thanks for your review and precious comments and advises. Specific responses are presented below:
>
> **Response to Question 1:**
>
> Thank you for your valuable comments. In the revised version, we will update Figure 3 by carefully adjusting the layout to make the expression of Figure 3 more aesthetically pleasing and clear.
>
> **Response to Question 2:**
>
> Thank you for identifying the mathematical notations that need clarification. In Section 3.2, we will add clear explanations for parameters k and n in the DFT formulation: "where k ∈ [0, T/2] is the frequency index in the transformed domain (representing frequencies from zero to Nyquist frequency) and n ∈ [0, T-1] is the time step index in the original signal." Additionally, for Equation (5), we will update the upper limit of the summation from T-1 to T/2 to correctly reflect our implementation. For Section 4.5, we will refine the optimization objective by removing the summation parameter S to present a cleaner formulation of the loss function, focusing on the core squared loss between prediction and ground truth.
>
> **Response to Other Comments Or Suggestions:**
>
> Thank you for your valuable comments. In the revised version, we will correct the errors in the quotation marks and thoroughly review the entire text to ensure that such issues do not recur.

---

> > ### Comment · Reviewer_LJCA · 2025-04-02
> >
> > Thank you for the clear responses. It’s clear the other reviewers also agree this work merits publication. I have no further concerns and am fully confident in recommending acceptance.

---

### Official Review · Reviewer_RXr7 · 2025-03-12

**Overall Recommendation:** 3

**Summary:**

This article investigates time series tasks and proposes a method called CFPT. Its main idea is to improve prediction results by capturing the complex relationships between different frequency components. The paper is written very clearly and the proposed modules have good motivation.

## Update after rebuttal

> I've read the rebuttal and other reviewers' comments, my final rating is weak accept. The reason why I give this score is that although the authors have already responded to most of the questions, I feel that the description of the CFI module is still not detailed enough and needs further elaboration.

**Claims And Evidence:**

Yes, I believe that the paper is supported by clear and convincing evidence.

**Essential References Not Discussed:**

N/A

**Experimental Designs Or Analyses:**

Yes, the experiment is reasonable. Ablation research, hyperparameter sensitivity analysis, efficiency evaluation, and performance visualization can validate the performance of the model.

**Methods And Evaluation Criteria:**

The article conducted extensive experiments on the seven commonly used datasets across diverse domains with comparing state of art baseline methods. The evaluation metrics used MSE and MAE. Overall, the method and evaluation criteria make sense for the time series forecasting problems.

**Other Comments Or Suggestions:**

The notation for coupling layers should be consistent throughout the paper. Figure 3 uses 'N' while the text uses 'L'.

**Other Strengths And Weaknesses:**

**Strengths**:
This work is significant for advancing time series forecasting by recognizing the crucial need to model both frequency component interactions and timestamp periodicities, the aspects overlooked in current methods. And the experimental results are reliable, while the evaluation demonstrates the method's strong performance through comprehensive evaluations in diverse forecasting scenarios under fair and rigorous evaluation protocols.

**Weaknesses**:
1. The introduction to Section 5 (Experiments) at its beginning could be more comprehensive. It is encouraged to briefly outline all experimental components, including the Ablation Study, Hyperparameter Analysis, and Visualization sections.
2. In addition to the current explanation, it is encouraged to provide further details on the splitting operation in the CFI branch.
3. In Section 3.3 (Instance Normalization), the symbol for the predicted normalized data in Equation 7 should be clarified.
4. In the Timestamp Hierarchical Processing (THP) section, the paper mentions that "each timestamp component is normalized to [-0.5, 0.5] through carefully designed transformations that preserve their cyclic characteristics", but doesn't specify the actual transformations used or explain how they maintain the cyclical nature of the temporal features.

**Questions For Authors:**

Based on the Weaknesses, I have the following questions:
(1)	Could you expand the introduction to Section 5 (Experiments) to include a brief overview of all experimental components?
(2)	How is the splitting operation in the CFI branch performed? Clarifying these details would enhance the reproducibility and understanding of the methodology.
(3)	What does the symbol in Equation 7 represent in the context of normalized data? Clarifying its meaning would enhance the understanding of the instance normalization process.

**Relation To Broader Scientific Literature:**

The paper advances the time series forecasting field, as evidenced by their related work discussion. In frequency-domain time series modeling, the proposed cross-frequency interaction mechanism represents an advancement over existing methods. In timestamp modeling research, the periodic-aware timestamp modeling provides a novel perspective compared to the existing methods that show limited effectiveness in practice.

**Theoretical Claims:**

N/A

---

> ### Author Rebuttal · Authors · 2025-03-31
>
> Many thanks for your review and precious comments and advises. Specific responses are presented below:
>
> **Response to Question 1:**
>
> Thank you for suggesting a more comprehensive introduction to Section 5. We will add the following sentence at the end of the opening paragraph: "Furthermore, we perform detailed ablation studies on both CFI and PTM branches, analyze hyperparameter sensitivity and computational efficiency, and visualize prediction results to demonstrate the effectiveness of our framework."
>
> **Response to Question 2:**
> Thank you for pointing out the need for more details on the splitting operation in the CFI branch. In Section 4.2 under "Frequency Recombination," we will replace the current text:
> "Specifically, we get low-frequency components$\hat{R}_L, \hat{I}_L \in \mathbb{R}^{N \times w'}$ and high-frequency components$\hat{R}_H, \hat{I}_H \in \mathbb{R}^{N \times w'}$."
> With this more detailed description:
> "Specifically, we split $\hat{g}_L \in \mathbb{R}^{N \times 2w'}$  to obtain low-frequency components $\hat{R}_L, \hat{I}_L \in \mathbb{R}^{N \times w'}$ and split $\hat{g}_H \in \mathbb{R}^{N \times 2w'}$ to obtain high-frequency components $\hat{R}_H, \hat{I}_H \in \mathbb{R}^{N \times w'}$, where the first $w'$ features represent the real parts and the remaining $w'$ features represent the imaginary parts. This splitting operation exactly reverses the concatenation performed in the initial feature processing stage, ensuring mathematical consistency when reconstructing complex numbers for the IDFT process."
>
> **Response to Question 3:**
>
> Thank you for your valuable comments. In the revised version, we will add an explanatory statement of the "predicted normalized data"($\hat X_{t + 1:t + \tau }^{norm}$).
>
> **Response to Other Comments Or Suggestions:**
>
> Thank you for your valuable comments. In the revised version, we will ensure consistency by using 'L' throughout the paper, including in Figure 3, to avoid any confusion for readers.

---

### Decision · Program_Chairs · 2025-05-01

**Decision:**

Accept (poster)

**Comment:**

**Summary:** This paper proposes CFPT, a novel architecture for long-term time series forecasting that explicitly models (1) cross-frequency interactions via a frequency-domain coupling mechanism, and (2) timestamp periodicity through a periodic-aware timestamp modeling branch using 2D CNN. The proposed method is evaluated on several real-world datasets across multiple prediction horizons and compared with eight recent state-of-the-art models. Extensive ablations and efficiency analyses demonstrate the utility and robustness of the proposed components.

**Decision:** All reviewers rated the paper positively and praised its novelty and solid empirical evaluation. The proposed cross-frequency interaction mechanism offers a fresh perspective beyond existing frequency-domain methods, and the periodic timestamp modeling adds a unique contribution to timestamp representation in time series forecasting. Minor concerns regarding clarity of figures, notation, and implementation details were effectively addressed in the rebuttal. As such, I recommend acceptance.